# Early Origins of Chronic Obstructive Pulmonary Disease: Prenatal and Early Life Risk Factors

**DOI:** 10.3390/ijerph20032294

**Published:** 2023-01-27

**Authors:** Michela Deolmi, Nicola Mattia Decarolis, Matteo Motta, Heidi Makrinioti, Valentina Fainardi, Giovanna Pisi, Susanna Esposito

**Affiliations:** 1Pediatric Clinic, Department of Medicine and Surgery, University of Parma, Via Gramsci 14, 43124 Parma, Italy; 2Department of Emergency Medicine, Massachusetts General Hospital, Harvard Medical School, Boston, MA 01451, USA; 3Cystic Fibrosis Unit, Pediatric Clinic, Az. Ospedaliera-Universitaria di Parma, Via Gramsci 14, 43124 Parma, Italy

**Keywords:** COPD, early life, lung function, lung trajectory, chronic obstructive pulmonary disease, prematurity, asthma

## Abstract

The main risk factor for chronic obstructive pulmonary disease (COPD) is active smoking. However, a considerable amount of people with COPD never smoked, and increasing evidence suggests that adult lung disease can have its origins in prenatal and early life. This article reviews some of the factors that can potentially affect lung development and lung function trajectories throughout the lifespan from genetics and prematurity to respiratory tract infections and childhood asthma. Maternal smoking and air pollution exposure were also analyzed among the environmental factors. The adoption of preventive strategies to avoid these risk factors since the prenatal period may be crucial to prevent, delay the onset or modify the progression of COPD lung disease throughout life.

## 1. Introduction

Chronic obstructive pulmonary disease (COPD) is the most prevalent non-communicable respiratory disease and the third leading cause of death worldwide. Based on recent epidemiological data, in 2021, about 300 million people, (i.e., 4% of the world’s population), suffered from COPD with an estimated number of deaths of 3.2 million per year [1,2]. The positive news is that COPD is often a preventable disease, as by avoiding smoking, 70% of cases are prevented. However, 30% of cases of COPD cannot be prevented by smoking avoidance [3]. The BOLD study found a different distribution of COPD severity among never smokers and smokers, according to GOLD stages. Among never smokers, 12% fulfilled the criteria for GOLD stage I or higher while among smokers this percentage increased to 23% [3]. Furthermore, among patients with COPD GOLD stage II+, the proportion of severe (GOLD stage III) and very severe (GOLD stage IV) airway obstruction was significantly lower in never smokers (5.9% vs 14.1) [3].

At a clinical symptoms level, COPD is characterized by persistent respiratory symptoms and airflow limitation due to airway and/or alveolar abnormalities caused by exposure to noxious particles or gases and is influenced by host factors, including abnormal lung development [1]. The main risk factors for COPD development consist of environmental variables (e.g., exposure to smoke, traffic-related air pollution, climatic changes). However, there is a plethora of other than environmental risk factors associated with COPD development. These factors vary from genetic, infectious, and inflammatory factors to social factors associated with lifestyle changes [2,3]. Most of the data about the pathological characteristics of COPD come from smokers and include an amplified inflammatory response to chronic irritants and morphological changes due to repeated injury and repair that persist even after smoking cessation. In non-smokers, other factors, such as systemic inflammation, genetic predisposition, oxidative stress, and changes in the lung microbiome may play a role [1].

Although COPD is a disease that typically develops in adulthood, there is increasing evidence about the importance of the first 1000 days of life that include the nine months of gestation and the first two year of life. During this period, there is a rapid development of tissues and organs, meaning that reduced quality or quantity of nutrients together with epigenetic influence and detrimental exposures may lead to a permanent organ remodeling. Therefore, several environmental determinants for non-communicable disease development can be tracked to the first 1000 days of life [4,5]. Specifically, for chronic respiratory disease development, these environmental determinants include air pollution and smoking exposures, early life respiratory infections (e.g., pneumonia or bronchiolitis), childhood asthma, malnutrition, and preterm birth [6,7]. Here, we review the main factors occurring in early life and childhood affecting the development of COPD in adulthood. 

### 1.1. Behind the Scenes: The Development of the Respiratory System

The morphological phases of lung development in utero start at approximately week 3–4 post-conception, when the respiratory diverticulum appears, derived from the primitive gut tube. Lung morphogenesis is traditionally divided into five different stages [8] (Figure 1). The *embryonic* stage starts from conception and goes until the 6th week of gestation. This stage is marked by the formation of the lung bud, a laryngotracheal groove of the ventral foregut, and initial branching of presumptive airways. When the groove closes, the region of the developing hypopharynx and larynx are the only regions that remain attached to the foregut. The lung bud gives rise to the conducting airways and to the five primordial lung lobes (two left and three right). By 7 weeks’ gestation, the trachea and the segmental and subsegmental bronchi are evident. In the *pseudoglandular* stage (weeks 6–17 of gestation), the respiratory tree undergoes twelve to fourteen more generations of branching, resulting in the formation of bronchi, terminal bronchioles, and acinar tubules. In the *canalicular* stage (weeks 16–26 of gestation), terminal bronchioles become surrounded with an increase in vascularization, and the process sets up the differentiation of specialized cell types. Pulmonary surfactant, a surface-active complex of phospholipids and proteins, is produced by type II alveolar cells after 24 weeks of pregnancy. The surfactant is essential for increasing pulmonary compliance, preventing atelectasis at the end of expiration, and facilitating the recruitment of collapsed airways. In the *saccular* stage (weeks 26–36 of gestation), respiratory bronchioles give rise to a final generation of terminal branches, forming the terminal sacs (primitive alveoli) that are lined with alveolar cells type I (the gas exchange surface in the alveolus) and type II (the cells that respond to damage of the type I cells). Finally, the *alveolar* stage (after 36 weeks post-conception) is characterized by the maturation of the alveoli, a process that continues many years after birth. It is only between the 34th and 36th week of pregnancy that fetal lungs have enough surfactant to allow the alveoli to remain open.

As a consequence, preterm birth can interrupt the physiological in utero development of the respiratory system and also interfere with its post-natal growth. Furthermore, adverse exposures such as maternal exposure to smoking or other pollutants occurring in one or more of the five stages can affect the structural development of the airways. 

There is increasing evidence that lungs can partly continue their development by neoalveolarization throughout childhood and adolescence, suggesting that there is somehow the potential to recover from some of these early life insults [9].

### 1.2. How Genes Influence the Respiratory System and Lung Function through Life

Lung function, measured as the forced expiratory volume in 1 s (FEV_1_), forced vital capacity (FVC) or their ratio (FEV_1_/FVC) shows a hereditable pattern, related to specific genes, both in the general population [10] and in people with COPD [11]. As a consequence, family history for COPD is an independent risk factor for COPD development but also for frequent exacerbations and more severe disease [11].

Animal models highlight the pivotal role of certain genes in lung morphogenesis, including retinoic acid receptor-beta (RARB), transcription factor SOX5, matrix metalloproteinase (MMP), and tissue inhibitors of MMPs (TIMPs). RARB knockout mice present abnormal alveolar septation, SOX5 deficiency leads to delayed lung development, and a lack of MMP14 reduces the surface area and delays angiogenesis [11]. Genes such as AGER, GPR126, GSTCD, HTR4, THSD4, and TSN1 are associated with lower lung function in the general population while other ones, including CHRNA3/4/5/7, CHRNB3/4, HHIP, and FAM13A, are associated with COPD development and the severity of airway obstruction [12]. The real-life result of these genes depends partly on nicotine addiction and cigarette exposure with a risk correlated to the gene–environment interaction [11]. The most widely known genetic variant associated with emphysema, which is one of the main characteristics of COPD, is the Z allele on the α1-antitrypsin gene (SERPIN), both in homozygosis and heterozygosis [13]. The wild-type form of the SERPIN gene is the M allele, while the most frequent variants are the S and Z types. Mutations in the SERPIN gene cause alpha-1 antitrypsin (AAT) deficiency, resulting in a predisposition to develop liver disease and lung emphysema because there are not enough levels of circulating AAT to contrast the effect of proteolytic enzymes such as neutrophil elastase in the lungs, causing parenchymal destruction. Smoking and occupational exposure to pollutants aggravate the lung damage, leading to COPD development [14]. Moreover, in a Pi*ZZ subject, the presence of alpha1-antitrypsin polymers promotes inflammation with consequent impairment in small airways and the long-term development of COPD [15]. AAT deficit is an under-diagnosed condition, but according to a recent study, its prevalence among patients with COPD is 1.29% [95% confidence interval (CI) 0.93–1.74], mainly the Pi*ZZ type (0.92%, 95% CI 0.62–1.31) and less frequently the Pi*SZ type (0.37%, 95% CI 0.19–0.64) [16]. However, levels of AAT do not seem to exactly correlate with the severity of lung damage. Recent literature suggests that other biomarkers should be studied, for example, miR-320c, that inhibits SERPINA1 expression, and its high blood levels are associated with lung disease [17].

Recent genome-wide association studies (GWAS) demonstrated that single-nucleotide polymorphisms (SNPs) linked to lung function deficits overlap with SNPs associated with COPD development. This could indicate that genes involved in the physiological growth of the lung may also be involved in the origin of COPD [18]. For example, two SNPs (rs1828591 and rs13118928) near the Hedgehog-interacting protein (HIP) gene, which is involved in early lung development and response to injuries, were significantly related to COPD development (OR 0.80, 95% CI 0.72–0.91), and this correlation was directly linked with the number of cigarettes smoked in a lifetime [19]. Similarly, certain gene variants may predispose COPD-like phenotypes if the subject is exposed to tobacco smoke. Smad3 knockout mice present an increased risk of emphysema due to reduced alveolar vascular endothelial growth factor (VEGF) and increased alveolar cell apoptosis; this condition is aggravated by tobacco smoke exposure that enhances Smad2 protein expression and phosphorylation [20]. Polymorphisms of ADAM33 5′ end result in a predisposition to airway obstruction in preschool children and to COPD susceptibility later in life. The interaction between ADAM33 and smoking exposure seems to be associated with a higher risk of impaired lung function and asthma development by age 8 [21,22]. Genes involved in metabolism and the clearance of endogenous products derived from oxidative stress such as EPHX1, CYP1A1, and GSTT1 can modify the impact of smoke and air pollutant exposure (PM_2.5_ and polycyclic aromatic hydrocarbons) on respiratory exacerbations and seem to be related to the development of emphysema and COPD later in life [23,24,25,26].

Bronchopulmonary dysplasia (BPD), a disease typically associated with prematurity, can result in a predisposition to COPD later in life, and several studies have been carried out to identify common genetic features between the two diseases. BPD seems to be more frequent in patients with polymorphisms of the SPOCK2 gene (rs1245560), and this polymorphism can be found both in Caucasian (aOR 2.96, 95% CI 1.37–6.40) and African preterm newborns (aOR 4.87, 95% CI 1.88–12.63). In Caucasian subjects, rs1049269 is a known polymorphism associated with BPD development (OR, 3.21, 95% CI 1.51–6.82) [27]; however, a GWAS conducted in extremely low birthweight infants with BPD found no significant correlation between BPD and SNPs but found that multiple SNPs in adenosine deaminase, CD44, and other genes involved in pathways of lung development were more frequent in patients with BPD, with a significant increase in miR-219 and CD44 levels [28]. This finding was confirmed in a case control study [29]. In contrast, moderate to severe BPD seems to show a hereditable pattern in twins [30]. BPD is not associated with a specific or larger number of copy-number variants (CNVs) compared to healthy controls [31], but exome sequencing showed novel variants of NOS2, MMP1, CRP, LBP, and TLR genes in BPD subjects born preterm. However, the pathogenic role of these variants needs to be established [32]. In twins, patients with BPD may have an excess of rare variants in genes associated with the regulation of the wingless/integrated (WNT) signaling pathway, morphogenesis of embryonic epithelium, collagen fibril organization, host defense, and extracellular matrix breakdown [33]. Telomere length was associated with abnormal airflow, but no correlation has been found with BPD [27,34]. Several differences in DNA methylation have been reported between term and preterm human lungs, but it is not certain if these changes are determinative in BPD or COPD development [35]. Interestingly, different gene expressions in the chromatin remodeling pathway have been reported only in preterm infants who developed BPD [36]. Sex and ethnicity also seem to play a role in BPD development. Males are at higher risk of prematurity and BPD and possibly a delayed production of surfactant [37] while black ethnic groups show a lower risk of BPD [38] despite an increased risk of persistent wheeze in preschool children born with extremely low birthweight [39]. Despite the strong link between BPD and COPD development, at present, it is not known whether any specific COPD gene variants are more likely to be present in children with BPD. Figure 2 summarizes the genes that may be implicated in COPD development and the interactions between different conditions.

### 1.3. Lung Function Trajectory and Spirometric Patterns

The concept of lung function trajectory comes from the Global Lung Initiative (GLI). GLI used 97,759 measurements performed in healthy non-smokers from 72 centers in 33 countries, aged between 2.5 and 95 years, to construct reference equations for description of the normal evolution of spirometry [40]. Three key features were identified to obtain a normal trajectory over life: lung function must not be afflicted by pre-birth adverse exposures (stage 1), needs to increase normally during childhood to reach a normal plateau at age 20 to 25 years (stage 2), and needs to decline in adulthood at a normal rate (stage 3).

#### 1.3.1. Stage 1, Pre-Birth Period

As already mentioned, lung development starts at week 3–4 of gestation and includes five stages of morphogenesis. Adverse environmental exposure occurring in one of these phases of development can interfere with the structural development of the airways and also result in airway hyperresponsiveness after birth [41].

#### 1.3.2. Stage 2, from Birth to Young Adult Age

During the first years of life, exposure to passive smoking [42] and outdoor pollution [43], early viral acute infections [44,45], and atopy [46,47] are considered important factors affecting lung growth and adult lung function. Several studies confirmed that lung function at age 4 to 6 years is dependent on neonatal lung function [48,49,50,51] and that lung function in young adulthood is dependent on school-age lung function [52,53,54,55]. During puberty, lung development differs in males and females. In males, lung and thoracic development, including lung volumes and pulmonary diffusing capacity, continues until the end of puberty while in females this occurs over a shorter period of time and is almost complete after menarche [56].

While there is evidence that alveolarization continues to adolescence with a sort of catch-up growth experienced by children born extremely preterm [9], the diffusing capacity of the lung seems reduced in puberty and early adulthood in this population compared with matched term-born controls with no signs of pubertal catch-up growth [57]. In clinical practice, the diffusing capacity is used to assess the severity and prognosis of COPD, and these findings may confirm that this test should be included in follow-up programs of extremely preterm babies.

An underlying key factor affecting the response of organisms to adverse exposures may be the airway microbiome. Early nasopharyngeal bacterial colonization has been associated with alteration in the physiological reaction of the immune system, a greater risk of wheezing, and impairment in response to respiratory infections [58,59]. On the other hand, bacterial and fungal diversity in the lung microbiome seems to be associated with a reduced risk of asthma. For example, Thorsen et al. examined the airway microbiota in a cohort of 700 children monitored for the development of asthma since birth: they concluded that microbial diversity in the airways at age one month were associated with asthma by age 6 years, and a higher relative abundance of *Veillonella* and *Prevotella* is furthermore associated with reduced TNF-α and IL-1β and increased CCL2 and CCL17, which itself is an independent predictor for asthma [60]. Furthermore, in other studies, alpha-diversity was reported to be higher in infants with wheezing and asthma at childhood [61,62].

Some studies demonstrated that during the first months of life, there was an enriched prevalence of pathogens such as *Haemophilus, Moraxella, Nisseria, Aerococcaceae, Staphylococcus,* and *Streptococcus* in children that developed wheezing and asthma. On the other hand, a decreased abundance of *Lachnospiraceae, Staphylococcaceae, Gemella, Veilonella, Leptotrichia, Granulicatella,* and *Dolosigranulum* is likewise more frequent in children with lung disease [61,63,64,65,66].

#### 1.3.3. Stage 3, Adult Life

Many patients with COPD show a normal rate of lung aging. However, asthma exacerbations, smoke and air pollution, severe pneumonia, concomitant diseases, and effective treatment of airway inflammation may accelerate the pulmonary function decline [67,68].

According to the literature, four patterns of airway growth can occur: (1) normal growth and normal decline, usually starting after the age of 18 years; (2) normal growth with early decline; (3) reduced growth with normal decline; (4) reduced growth with early decline (Figure 3). The latter two patterns are associated with the development of COPD [69,70,71]. Unfortunately, all of these patterns can be clearly recognized only retrospectively, but the potential early identification of these patients can be crucial.

The complex relationship of obstructive and restrictive spirometric phenotypes with early-life risk factors from childhood to young adulthood was further explored in a wide population study including several European cohorts [72]. The obstructive pattern observed from school age to young adulthood was significantly associated with maternal smoking during pregnancy (aOR 1.16, 95% CI 1.01–1.35), preterm birth (aOR 1.84, 1.27–2.66), diagnosis of asthma (aOR 2.55, 95% CI 2.14–3.04), and a family history of asthma (aOR 1.44, 95% CI 1.25–1.66).

### 1.4. Dysanapsis

The term “dysanapsis” indicates a disproportionate growth between lung parenchyma and airway caliber [73]. The phenomenon of “dysanaptic lung growth” probably starts at birth or even antenatally. A greater lung volume improvement corresponds to a smaller growth of the airway size, resulting in a higher FVC than FEV_1_ and a lower than expected FEV_1_/FVC ratio [74]. 

Airway remodeling has been suggested as one of the reasons why FEV_1_ can be lowered. This may be related to both environmental insults such as air pollution and smoking, as well as severe disease exacerbations, BPD or genetic factors. 

For example, children with BPD have evidence of a reduction in expiratory flow rates with air trapping. This finding might be secondary to small airway narrowing, a reduction in alveolar number, and a reduction in elastic recoil [75,76].

In a recent population-based prospective cohort study, Sonnenschein-van der Voort et al. estimated individual growth trajectories from birth until 10 years of age and concluded that faster weight growth in early childhood was associated with asthma and bronchial hyperresponsiveness [77]. A higher body mass index (BMI) correlated negatively with the FEV_1_/FVC ratio from childhood to young adulthood and has been therefore associated with an increased likelihood of the obstructive phenotype [78]. BMI gain during early childhood seems to have greater influence on lung volume than airway growth, which may lead to airway dysanapsis [73,79].

#### The “Window of Susceptibility” That Increases the Risk of COPD

Between the second part of gestation and the first years of life, certain exposures can cause structural or functional abnormality in lung development and therefore in respiratory function. This period has been called the “window of susceptibility” [80]. Depending on the type of exposure and genetic predisposition, this susceptibility can result in different degrees of abnormalities and eventually in a higher risk of COPD development.

The assessment of respiratory function by lung function tests, particularly with spirometry, allows an indirect measurement of lung growth. Lung volumes increase from childhood to young adulthood with maximum values reached at 20–25 years of age, remaining on a plateau for around five years, and then gradually decreasing with aging. This process is called the trajectory of lung function and starts early in life. Lung function at birth and in preschool age can predict the maximum lung volume a person will reach [49], and low values represent a risk factor for persistent airflow obstruction in adults [81].

Early or accelerated decline in lung function and eventually the development of COPD can be the manifestation of different exposures combined with genetic predisposition and not only the result of tobacco smoke as suggested by previous studies [82]. 

The natural history towards COPD can present different features in each individual. Lange et al. stratified 657 subjects according to lung function (FEV_1_ ≥ 80% or <80% of the predicted value) at approximately 40 years. Of those with an FEV_1_ <80% before age 40 years, 26% had developed COPD during the observational period, whereas of those with an FEV_1_ ≥ 80%, only 7% developed COPD within the same time period due to an accelerated rate of decline in pulmonary function [71]. Wang et al. conducted a meta-analysis that included 49,334 subjects from 14 population-based cohorts in different age groups with the aim of studying how obstructive or restrictive spirometric patterns occur over time. The authors concluded that these pathologic lung function phenotypes seem to be relatively prevalent during childhood, supporting the early origins of respiratory decline seen in COPD [77].

The logical conclusion is that COPD risk is mainly determined within the first few decades of life where many factors can play a role in the development of respiratory function and in the reach of maximal (or “plateau”) FEV_1_ [83].

### 1.5. Intrauterine Growth Restriction (IUGR), Preterm Birth, and Low BirthWeight

In 1986, David Barker, an English physician, hypothesized the theory of fetal origins of adult disease founded on the concept that the period of gestation had significant impacts on the developmental health. According to this theory, poor nutrition when tissues and organs are shaping in intrauterine life leads to a permanent remodeling, in particular seen as intrauterine growth restriction, that may be responsible for the development of some non-communicable disease later in life.

Several studies showed a positive linear trend between birthweight and lung function in adulthood measured as FEV_1_ and FVC [84,85,86], and this is particularly true for infants born small for gestational age (SGA) [87,88]. A wide cross-sectional study conducted in England in 2257 women found a positive association between birthweight and lung function between 60 and 79 years. After adjustments for age, smoking, and height, FEV_1_ increased by 48 ml for every kilogram of birthweight (95% CI 0.026 to 0.070) [85]. Similarly, in a longitudinal study including 2957 subjects, lower birthweight was associated with lower FEV_1_, suggesting a decreased trajectory of lung function later in life [86]. Broström and colleagues followed over time a Swedish cohort of newborns born preterm (before 35 weeks of gestational age) or born with low birthweight (lower than 2000 g for girls and lower than 2100 g for boys) and found that, compared to controls born at term and with a normal birthweight, women showed a higher risk of developing obstructive pulmonary disease such as asthma or COPD while no difference was found in men [89]. On the other hand, an English study demonstrated that men aged 59 to 70 years had an increase of 60 ml in FEV_1_ for every 450 g increase in birthweight regardless of smoking habits [90]. Consistent results were obtained in childhood in a cohort of 2,036 children aged 5 to 11 years where lower birthweight was associated with lower FVC (b = 0.475, 95% CI 0.181–0.769) and FEV_1_ (b = 0.502, 95% CI 0.204 to 0.800) after adjustment for gestational age and other confounding factors [91]. The Avon Longitudinal Study of Parents and Children assessed the lung function of children born at term at 8 and 9 years of age comparing those with or without IUGR. Children with IUGR had worse lung function measured as FEV_1_ (−0.198, 95% CI −0.294–−0.102), FVC (−0.131, 95% CI −0.227–−0.036), and FEF_25–75_ (−0.149, 95% CI −0.246–−0.053) that translated to a decrease of approximately 50 ml for FEV_1_, 40 ml for FVC, and 80 ml for FEF_25–75_. Interestingly, subjects of the IUGR group showing catch-up growth during infancy seemed to have better lung function [92]. A recent meta-analysis demonstrated a significant association between low birthweight and the increased probability of COPD development (three studies included; n = 5176, pooled aOR 1.58, 95% CI 1.08–2.32) but not with prematurity (three studies included; n = 4410, pooled aOR, 1.17, 95% CI 0.87–1.58) [6].

Preterm birth (defined as birth before 37 weeks of gestational age) occurs in 5 to 10% of all births. Despite the increasing survival rate, significant morbidity, especially respiratory tract complications, still affect short-term and long-term outcomes. In preterm infants, lung development is interrupted before its complete development, and the further exposure to oxygen, mechanical ventilation, oxidative stress, and infections can lead to BPD and other milder types of respiratory impairment even in late preterm infants or in those who did not need ventilation. However, regardless of BPD, preschool children born at 22–31 weeks of gestational age showed higher airway resistance and greater impedance in a cohort of 194 Italian children [93]. Similarly, Pelkonen and colleagues demonstrated that at 8 to 14 years of age, premature newborns had lower spirometric values compared to the controls born at term, suggesting that prematurity alone can negatively affect lung development [94]. Moreover, in the Avon Longitudinal Study cohort, infants born at 33–34 weeks of gestational age presented lower lung function values at 8–9 years of age compared to controls but in contrast with children born at 25–32 weeks of gestational age, showed some improvements in lung function by 14–17 years [95]. Interestingly, a recent systematic review revealed that children born moderate–late preterm had worse expiratory airflows than those born at term, suggesting that even a relatively late interruption of in utero lung development can affect lung function later in age [96].

In a wide population study including lung function studies of almost 25,000 subjects from early childhood to young adulthood, lower gestational age and smaller birthweight for gestational age were associated with lower FEV_1_. Furthermore, preterm birth and low birthweight were associated with an increased risk of childhood asthma [pooled odds ratio, 1.34 (95% CI 1.15–1.57), 1.32 (95% CI 1.07–1.62)] [97]. Preterm infants with a lower than normal lung function trajectory and failing to reach the normal peak at 20–25 years of age may be at higher risk of developing a COPD-like phenotype later in life [98,99,100,101,102]. As already demonstrated by Peto and Fletcher in 1977, not all patients with COPD experience an excess loss of lung function [82]. Chronic airflow limitation can develop through different trajectories, some originating from reduced development of lung function in childhood and adolescence [71].

### 1.6. Bronchopulmonary Dysplasia

BPD is one of the short- and long-term complications of prematurity presenting with a frequency inversely correlated with gestational age [103]. Depending on countries and definition, BPD affects between 10% and 89% of preterm infants and about 45% of those born at less than 29 weeks of gestation [104,105,106]. BPD had been initially described as the result of an aggressive mechanical ventilator approach in terms of peak pressures and oxygen concentrations on a relatively mature lung lacking surfactant (i.e., ≥32 weeks of gestation) [107] that resulted in the need for oxygen for at least 28 days after birth [108]. However, the pathogenesis of the disease is more complex. The impaired lung development resulting in BPD likely begins before delivery in many infants [109]. Despite the introduction of early rescue surfactant treatment, antenatal glucocorticoids and more gentle ventilation, injures occurring in the premature lung can still result in BPD development. From a histological point of view, BPD is characterized by the disruption of alveolarization with simplified alveoli and reduced septation, as well as dysmorphic pulmonary microvessel growth with a decreased surface area for alveolar-capillary gas exchange [110,111,112]. In addition, mucus gland hyperplasia, epithelial oedema, and smooth muscle cell proliferation may impair small airway development, promoting bronchoconstriction [113]. Since alveolarization is a prolonged process, starting in utero and resulting in a 20-fold increase in the surface area from birth (0–50 million alveoli) to adulthood (>300 million alveoli), all of the prenatal and postnatal factors interrupting this crucial phase of lung development may contribute to BPD development. A list of these factors is shown in Table 1.

Extremely low gestational age newborns (i.e., <28 weeks of gestation) are those at the highest risk of developing BPD. They are born before alveoli precursors’ formation (saccules, alveolar ducts, alveolar air sacs) and increase of the capillary bed [123,124] and before a sufficient amount of surfactant has been produced [12,110]. 

Children with BPD have a hospitalization rate up to 50% during the first two years of life [125,126]. Respiratory syncytial virus and rhinovirus may increase the risk of hospital admissions in this age group, as they frequently precipitate pulmonary exacerbations [127]. Children with BPD hospitalized for common respiratory viral infections early in life have worse lung function in later childhood [128,129]. Furthermore, impaired exercise tolerance is a common finding in BPD survivors [130,131]. At approximately 52 weeks postmenstrual age, about half of the subjects with severe BPD, showed an obstructive pattern with a significant response to bronchodilators in most cases [132]. Up to two years of age, reduced expiratory flows measured by the raised volume rapid thoraco-abdominal compression technique and the measurement of the maximal expiratory flow at functional residual capacity (Vmax_FRC_) have been demonstrated in preterm infants with and without BPD [133,134]. Compared to non-BPD subjects, BPD children show substantial impairments in airflow during childhood and adolescence with lower FEV_1_ values significantly associated with BPD severity. Doyle et al. demonstrated a reduction in all lung function variables in young adults with BPD with an obstructive pattern that deteriorated most between 8 and 18 years old, affecting the reach of the maximally attained FEV_1_ [135]. A meta-analysis of 86 studies involving children and young adults born prematurely concluded that preterm children with BPD had the lowest FEV_1_ with a deficit of up to 16% compared to term-born controls [136]. Simpson et al. found that children born preterm manifested a decline in spirometric values from 4 to 12 years of age with the steepest and fastest trajectory in children with BPD [137,138]. Preterm infants with and without BPD show a progressive decrease in FEV1 and the FEV_1_/FVC ratio and do not catch up to the lung function of their matched term peers, failing to reach the normal peak at 20–25 years of age. These findings suggest a route towards the development of the early onset of chronic airway obstruction [100,102,139]. 

Two factors contributing to impaired lung function and to the increased risk of COPD in children with BPD may be the nonsynchronous increase in lung size and airway caliber, such as dysanapsis [110,140], and the persistence of chronic airway inflammation, as demonstrated by studies on exhaled breath condensate [141].

## 2. Environmental Exposures

### 2.1. Tobacco: In Utero and Early Life Exposure

Active smoking and second-hand smoke exposure are already known as important independent risk factors for COPD development [142,143]. 

Increasing evidence suggests that intrauterine smoking exposure is associated with impaired lung development, abnormal lung function during childhood, and higher risk for recurrent wheeze and asthma development [86,144,145,146,147,148,149]. The European Community Respiratory Health Survey that included 18,922 subjects aged 20 to 44 years showed that in males, paternal smoke exposure during childhood was associated with wheezing symptoms with a dose-dependent effect and with a reduction in the FEV_1_/FVC ratio (−0.3%, 95% CI −0.6–0). Similar findings were obtained in women where maternal smoking exposure was associated with wheezing and lower values of FEV_1_ and the FEV_1_/FVC ratio (−0.6%, 95% CI −0.9–−0.3); no association was found in women with paternal smoking exposure [147]. Moreover, exposure to maternal smoking at the time of delivery seems to decrease the FEV_1_/FVC ratio and increase the risk of hospitalization or death from COPD [150]. Håberg and colleagues analyzed the Norwegian Mother and Child Cohort and, thanks to questionnaires administered at different times of pregnancy and puerperium, found that both intrauterine smoke exposure and early life paternal smoke exposure were independent risk factors for wheezing symptoms and respiratory infections [151].

Proposed mechanisms of damage related to intrauterine smoke exposure include a shortening of telomere length or DNA methylation and reduced expression of genes involved in lung development, but also the direct effect of smoking on respiratory mucosa [6,152,153,154,155,156,157]. Notably, some studies in animals suggest that these effects may be mediated by nicotine, meaning that e-cigarettes containing nicotine may also be harmful [12].

Tobacco smoke exposure in early life is independently associated with recurrent wheezing and asthma development as demonstrated by a pooled analysis performed on 53,879 subjects [158]. In the MESA (Multi-Ethnic Study of Atherosclerosis) cohort, non-smokers subjects (n = 1781) with early life exposure to tobacco smoke were at a higher risk of early development of emphysema even after the adjustment for air pollution exposure and exposure to tobacco smoke in adulthood. Similar results were recorded in lung CT-scans of non-smoking adults where the percentage of emphysema was significantly associated with early life smoke exposure. The underlying mechanism can be the mechanical stress caused by smoke particles to the small airway walls during childhood that eventually result in permanent damage later in life [159]. A case-control study involving the Bergen COPD Cohort (433 patients with COPD vs. 325 healthy control subjects) showed that early tobacco smoke exposure was associated with higher risk of COPD among women (OR 1.9, 95% CI 1.0–3.7) and with more respiratory symptoms among men (dyspnea OR 1.5, 95% CI 0.9–2.5 and cough OR 1.7, 95% CI 1.1–2.6) [160].

Taking together these results, we support the hypothesis that tobacco exposure in utero and in early life increases the risk of the development of an obstructive respiratory phenotype and therefore of COPD.

### 2.2. Air Pollution

Air pollutants [i.e., nitrogen oxide (NO), sulfur dioxide (SO2), carbon monoxide (CO), volatile organic compounds (VOC), and particulate matter (PM)], particularly particles smaller than 2.5 mm, are associated with respiratory disease development through various mediation pathways. These mediation pathways include systemic and non-systemic inflammatory processes, including oxidative stress and the impact on the epigenetic control of the lung [161,162]. These changes can therefore result in an increased risk for COPD development in adulthood [163,164,165].

In regard to prenatal air pollution exposures, maternal exposures are associated with IUGR, preterm delivery, low birthweight, and impaired lung function [166,167,168,169].

Exposure to air pollutants in early life is associated with impaired lung function development, especially in children with asthma [170]. Air pollution associated with traffic (TRAP) during childhood has been related to lower FEV_1_ and risk of COPD [171].

In addition to TRAP exposures, indoor air pollution exposures are important sources of pollutants. For example, indoor bio-mass combustion derived from the use of solid fuels or wood combustion for cooking and heating has been linked to accelerated lung function decline, a higher prevalence of COPD and higher all-cause and respiratory cause-specific mortality [172,173]. PM_2.5_ released from candle burning was associated with prevalent impaired FEV_1_ in a few observational studies [174,175].

## 3. Respiratory Infections in Early Life

Early-life respiratory viral infections have been linked to an increased risk for asthma and COPD development. Specific viral agents, e.g., rhinovirus and respiratory syncytial virus, have a stronger link to later asthma development [44,176]. This is associated with either genetic susceptibility to specific respiratory viruses (e.g., 17q21 locus in association with rhinovirus-induced wheezing), which has been linked to asthma development, or co-existing allergic sensitization [177].

However, it is unclear whether a respiratory viral infection comes as the “first hit” in the asthma development pathway. Cohort studies have shown that impaired lung function can precede the first wheezing episode [45], and immunological abnormalities have been identified in the cord blood of babies who subsequently develop wheezing lower respiratory tract illness [178]. Most of the studies agree that childhood infections and asthma are related to COPD development later in life [179,180,181,182]. 

Hayden et al. investigated the association between childhood pneumonia and COPD in adult former and current smokers (n = 10,192), finding that a history of pneumonia during childhood was associated with chronic bronchitis and COPD development [181]. In a large prospective cohort study conducted by Tagiyeva et al., childhood wheezy bronchitis and childhood asthma were significantly associated with an increased risk of COPD at the age of 60–65 years [182]. Similarly, in a longitudinal study conducted by Chan et al., the FEV_1_/FVC ratio was significantly lower in adults with a radiologically ascertained pneumonia before 3 years of age than in those with lower respiratory tract illness but without pneumonia [183].

Both infections in early life and a certain degree of lung function impairment present before the infection itself can contribute to the development of COPD.

## 4. Asthma in Childhood

Children with persistent asthma are at higher risk for fixed airflow obstruction and eventually early-onset COPD. Patients with persistent asthma can present reduced airway growth with more severe asthma symptoms, higher resistance, and airway hyper-responsiveness. Physiologic studies showed that subjects with the lowest FEV_1_/FVC ratio have greater airway hyper-reactivity [184].

A recent meta-analysis showed that children with asthma have 3.45-fold increased probability to develop COPD than children without asthma (pooled aOR 3.45, 95% CI 2.37–5.02) [6]. Already in 2012, a retrospective study conducted by Shirtcliffe and colleagues highlighted that 18.9% of included patients with COPD (n = 116), had a history of childhood asthma [181]. A case control study in the Japanese population showed that patients aged 50–75 years suffering from COPD had more frequently a history of childhood asthma than controls (6.3% vs 2.4%, *p* = 0.015) and that childhood asthma was significantly associated with the risk of developing COPD in adulthood (aOR: 3.32, 95% CI 1.05–10.45) [185]. In 2014, Tai et al. published a prospective study based on the Melbourne asthma cohort that enrolled patients at 7 years of age and followed them up until the age of 50 (n = 375). The authors found that children with severe asthma (onset before 3 years of age and FEV_1_/FVC at 10 years lower than 70% associated with persistent symptoms) had 32 times higher risk of developing COPD compared to controls [179].

In another prospective cohort study, Tagiyeva and colleagues pointed out that wheezy bronchitis (OR 1.81; 95% CI 1.12–2.91) and childhood asthma (OR, 6.37; 95% CI 3.73–10.94) were associated with increased risk of COPD in patients followed-up until 60–65 years of age [182]. The strongest association between asthma and COPD was reported in a study conducted in Tunisia (OR 10.62, 95% CI 2.90–38.94, *p* < 0.01) [186]. Similar results were obtained in a Canadian cohort where a history of physician-diagnosed asthma was significantly related to COPD prevalence (OR 3.30, 95% CI 2.42–4.49) and in the BAMSE birth cohort [72,187].

Although still controversial, one of the proposed mechanisms includes airway inflammation and remodeling that can lead to irreversible airway obstruction and reduced FEV_1_ [188]. In a study including 9896 participants with an age range between 35 and 60 years, subjects with childhood–adulthood asthma had a higher prevalence of airway obstruction (16.8%) defined as FEV_1_/FVC < 70% compared to those with remitted childhood asthma (5.2%) and adult-onset asthma (14.4%). Interestingly, after multivariate logistic regression, clinically remitted childhood asthma appeared to be independently associated with airflow obstruction in middle-aged adults [189].

A recent study cohort of 3,290 children demonstrated that subjects with asthma-like symptoms in early childhood showed reduced predicted FEV_1_ percentage and FEV_1_-/FVC ratio at age 50 years [−3.36% (95% CI −5.47 to −1.24) and −1.28 (95% CI −2.17 to −0.38), respectively] with an increased risk of COPD diagnosis at adult age (OR 1.96, 95% CI 1.13–3.34)] [190].

Comorbidities affecting lung function trajectories have been assessed in a prospective cohort study where COPD was most strongly associated with early-onset persistent asthma and allergy trajectory (OR 5.3, 95% CI 3.2–8.6]) and also with late-onset asthma and allergy trajectory (OR 3.8, 95% CI 2.4–6.1]) [50].

Growing evidence suggests that asthma can overlap with certain COPD phenotypes resulting in a disease termed “asthma/COPD overlap syndrome”. These patients show persistent airflow obstruction and a history of asthma during childhood. The hypothesis is that in patients with poorly controlled asthma, the rate of FEV_1_ decline can be more rapid than in patients with good control [191,192]. Interestingly, Bui suggested that childhood lung function can predict COPD and asthma/COPD overlap syndrome, showing that adults with asthma/COPD overlap syndrome had a lower FEV_1_/FVC ratio not only at the age of 45 but also at the age of 7 years [193].

## 5. Conclusions

COPD is a chronic respiratory condition that starts early in life. There are studies showing that its origins can be located even earlier, in the prenatal period. The causal pathway from birth until COPD development is mediated by several factors (Figure 4). Some factors act as confounders and others as effect modifiers. However, the inter-relationship between these factors in regard to COPD development is not fully understood. The role of environmentally induced changes in epigenetic control are key in COPD pathogenesis. Other factors, e.g., genetic susceptibility, respiratory infections, allergies, and history of prematurity, also play a significant role in COPD pathogenesis. Smoking is the most strongly implicated exposure. However, 30% of COPD cases develop in non-smokers. Identifying sensitive predictors for COPD development can help us design effective prevention strategies in the future. Since at present COPD is treated with palliative and symptomatic treatments, primary prevention to avoid detrimental exposures is crucial to maintain regular lung growth and reduce the prevalence of this irreversible non-communicable disease.

## Figures and Tables

**Figure 1 ijerph-20-02294-f001:**
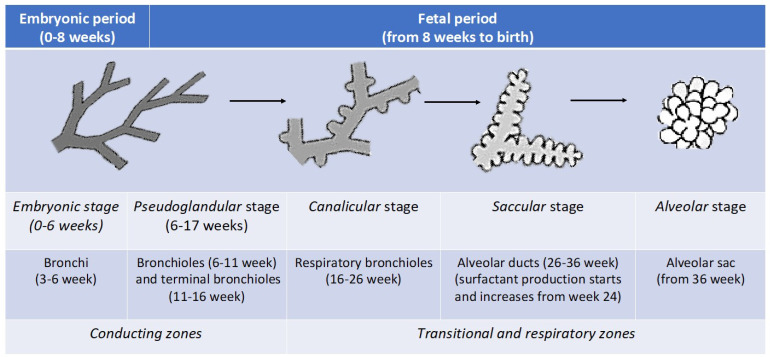
Five different stages of lung development.

**Figure 2 ijerph-20-02294-f002:**
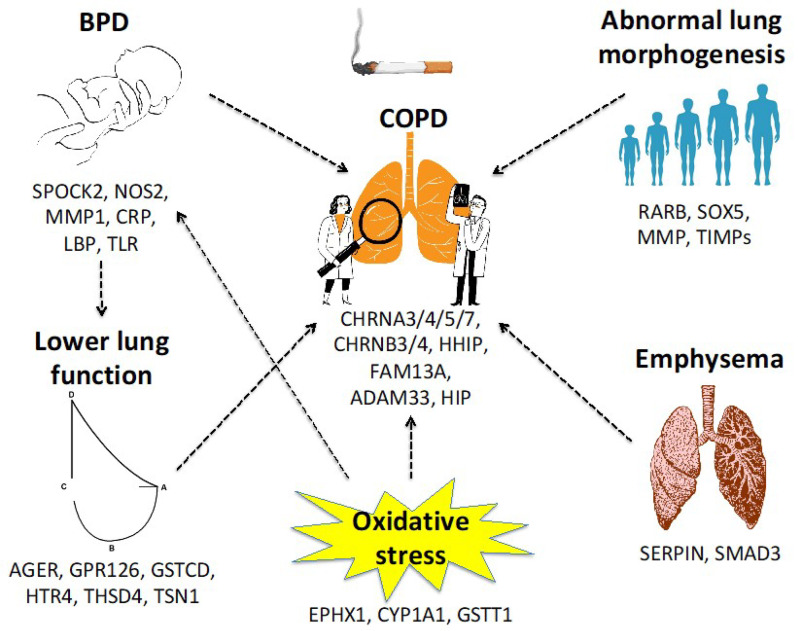
Genes that may be involved in COPD development directly or indirectly through other conditions.

**Figure 3 ijerph-20-02294-f003:**
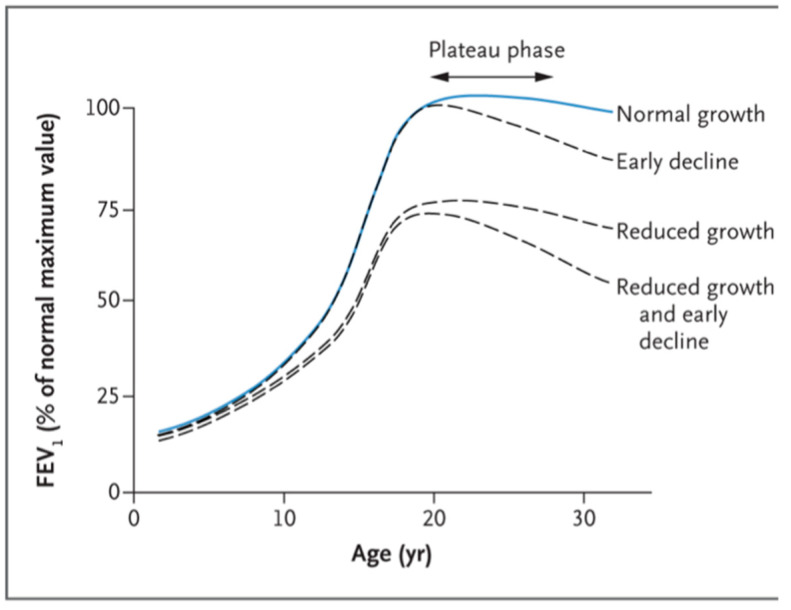
Longitudinal lung-function trajectories reproduced with permission from McGeachie et al. [70].

**Figure 4 ijerph-20-02294-f004:**
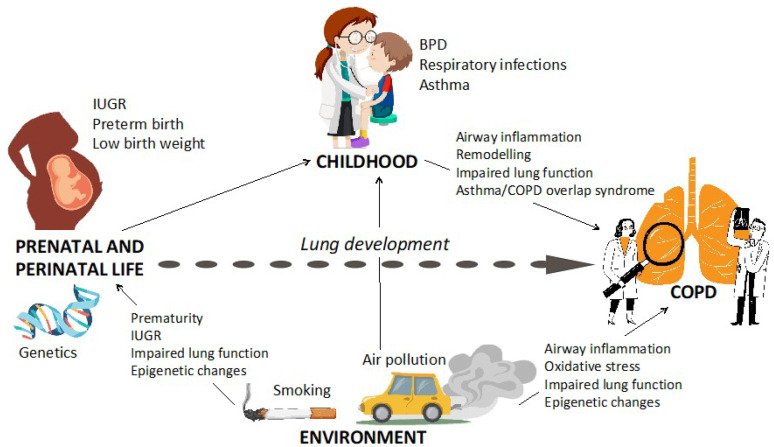
Factors in prenatal and perinatal life, the environment, and childhood affecting the risk of developing COPD in adulthood. IUGR, intrauterine growth restriction; BPD, bronchopulmonary dysplasia; COPD, chronic obstructive lung disease.

**Table 1 ijerph-20-02294-t001:** Risk factors associated with BPD development.

Risk Factors Associated with BPD
Gestational age, birthweight [98,99,100,101,102]
Mechanical trauma and oxygen toxicity related to ventilation [107,114,115,116]
Maternal smoking, hypertension, preeclampsia [117]
Prenatal infections [118]
Postnatal infections [119,120]
IUGR [121]
Poor nutrition [122]
Genetics [27,29,30]

## Data Availability

Not applicable.

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
