# Peer review of "Early Origins of Chronic Obstructive Pulmonary Disease: Prenatal and Early Life Risk Factors"

_ijerph, 2023, doi:10.3390/ijerph20032294_

Round 1

Reviewer 1 Report

Summary of knowledge of prenatal and early life factors leading to COPD in never smokers. The topic is highly relevant and the ms addresses a specific gap in the field of COPD. The expertise on lung function in development and its aberrations and on the influence of intrinsic/genetic and environmental factors for the development of COPD in never smokers. the conclusions are consistent. The references are appropriate. Tables and Figures are appropriate, I like the last figure which is like a graphical abstract.

This is a very well written review on prenatal and early life factors in the development of COPD.

Minor concerns

Please, include:

A) A § on comparability of COPD pathology in smokers and never smokers

B) A § on the comparability of the severity of the disease in smokers and never smokers (e.g. distribution of GOLD stages)

C) A simple introduction/definition of Dysanapsis

Author Response

A) A % on comparability of COPD pathology in smokers and never smokers

We thank the reviewer for this comment. We added in the first paragraph a sentence with these percentages.

B) A % on the comparability of the severity of the disease in smokers and never smokers (e.g. distribution of GOLD stages)

Thanks for this comment. We added in the first paragraph a sentence on this.

C) A simple introduction/definition of Dysanapsis

Good point, thank you. The definition has been added in the text.

Reviewer 2 Report

Very well organized paper looking at early origins of COPD.

It does need some editing with regard to verbiage ( phrases like "it has been shown", "It is clear", "it has been speculated" etc.  can be found throughout the paper - these and similar phrases need to be removed)

Author Response

It does need some editing with regard to verbiage ( phrases like "it has been shown", "It is clear", "it has been speculated" etc.  can be found throughout the paper - these and similar phrases need to be removed)

Thanks for the comment. The phrases have been changed accordingly.

Reviewer 3 Report

This is an interesting and well-written article that nicely summarizes our current understanding of the topic and is currently topical in both adult and paediatric respiratory function and measurement. 

The only comment I have that may need some attention is that the article is a little heavy in places and perhaps would be enhanced more by some additional figures. There are a few other suggestions listed below as well. 

The section starting at line 53 - there are some good diagrams available in the literature which summarize this section, perhaps more clearly showing the overall development of the lungs in the timescale of development. This would help to more visually capture lung development.

The section starting at line 87 - is there a way of visualizing the effects and interactions of genes diagrammatically? The current Figure 2 is a good visually appealing diagram - can something similar be undertaken for this section?

Line 175 - the word predictive could be changed to reference

Line 186 et sec - is there any evidence of development around puberty and whether this is affected by the topics discussed? 

Line 248 - will the readers be conversant with the use of z-scores? Is suspect that many may not, being more focused on %predicted. This may need expanding a little more - but I'm delighted to see their mention.

Line 268 - many will be familiar with the Fletcher-Peto diagram. It has great relevance to understanding what happens in adult respiratory practice and communication with patients regarding the rate of decline in their lung function. You might consider expanding on its relevance and, following on from Figure 1, how this might be more affected by early life events.

Line 365 - this references Table 2, but I cannot see Table 1

Line 553 - Figure 2 - there appear to be some errors in this diagram. Good diagram, though.

Author Response

The section starting at line 53 - there are some good diagrams available in the literature which summarize this section, perhaps more clearly showing the overall development of the lungs in the timescale of development. This would help to more visually capture lung development.

Thanks for this comment. We created a new figure summarizing lung development.

The section starting at line 87 - is there a way of visualizing the effects and interactions of genes diagrammatically? The current Figure 2 is a good visually appealing diagram - can something similar be undertaken for this section?

Thanks for the comment. We created a diagram highlighting the interactions between genes and specific conditions.

Line 175 - the word predictive could be changed to reference

Changed accordingly.

Line 186 et sec - is there any evidence of development around puberty and whether this is affected by the topics discussed? 

Thanks for the comment. We added a paragraph of lung development during puberty.

Line 248 - will the readers be conversant with the use of z-scores? Is suspect that many may not, being more focused on %predicted. This may need expanding a little more - but I'm delighted to see their mention.

Yes, thank you, z-scores might be difficult to interpret. We inserted the volume lost for each measurement. 

Line 268 - many will be familiar with the Fletcher-Peto diagram. It has great relevance to understanding what happens in adult respiratory practice and communication with patients regarding the rate of decline in their lung function. You might consider expanding on its relevance and, following on from Figure 1, how this might be more affected by early life events.

Thank you very much for this comment. We added a sentence with this reference.

Line 365 - this references Table 2, but I cannot see Table 1

Apologies. That was a mistake. We meant Table 1.

Line 553 - Figure 2 - there appear to be some errors in this diagram. Good diagram, though.

Apologies, can you please specify the errors. We checked for grammatical errors but we can not see them.